# Evaluation of Entomopathogenic Nematodes against Red Palm Weevil, *Rhynchophorus ferrugineus* (Olivier) (Coleoptera: Curculionidae)

**DOI:** 10.3390/insects13080733

**Published:** 2022-08-16

**Authors:** Gul Rehman, Muhammad Mamoon-ur-Rashid

**Affiliations:** Department of Entomology, Faculty of Agriculture, Gomal University, Dera Ismail Khan 29220, Pakistan

**Keywords:** date palm weevil, biological control, entomopathogenic nematodes, mortality

## Abstract

**Simple Summary:**

The red palm weevil is considered the most notorious pest for different species of palms around the globe. Due to the concealed nature of the red palm weevil, the use of biocontrol agents, especially EPNs, is considered most effective. Among biocontrol agents, entomopathogenic nematodes provide effective control of the different developmental stages of the red palm weevil. In the current investigation, the infective capabilities of four different species of entomopathogenic nematodes were investigated against larvae (5th and 6th instars), pupae enclosed in their cocoons, and adult red palm weevil (newly formed), under laboratory and field conditions. Our results indicated that the *S. carpocapsae* and *H. bacteriophora* were the most effective EPN species against different developmental stages of the red palm weevil under laboratory as well as field conditions. The larval stage of the red palm weevil was found to be the most susceptible to infection by EPNs compared with pupal and adult stages. From the present findings, we can infer that *S. carpocapsae* and *H. bacteriophora* are the most effective EPN strains and recommend their use in the sustainable management of the red palm weevil.

**Abstract:**

Entomopathogenic nematodes play a pivotal role as biocontrol agents for different species of insect pests, including the red palm weevil. In the current investigation, the infective capabilities of four species of entomopathogenic nematodes, including *Hetrerorhabditis bacteriophora*, *Steinernema feltiae*, *Steinernema glaseri*, and *Steinernema carpocapsae*, were evaluated against larvae, pupae, and adult red palm weevil under laboratory and field conditions. The pathogenic potential of selected nematode species was assessed based on dissection and adult emergence of weevils. Our results indicated that *S. carpocapsae* and *H. bacteriophora*, with a respective 94.68 and 92.68% infection rate, were the most effective EPN species against red palm weevil larvae. Focusing on adult emergence, the aforementioned EPNs were comparatively less pathogenic and resulted in 63.60 and 60.20% infested pupae, respectively. It is noted that adult emergence is the better option to evaluate the pathogenic potential of EPNs, compared with the dissection of insects. The *S. carpocapsae* was found to be most effective against the 6th instar larvae of the red palm weevil and caused 100% mortality at 240 h after treatment. On the other hand, *S. glaseri* and *S. feltiae* were found to be the least pathogenic and caused 70 and 76% mortality, respectively. All of the evaluated nematode species were found to be highly infective under field conditions. The *S. carpocapsae* was found to be the most pathogenic, causing 83.60% mortality of the red palm weevil. However, the tested nematodes were found most effective against larvae, followed by adult weevils, but their effect was minimal against the pupae of red palm weevils. Based on these findings, we conclude that the *S. carpocapsae* and *H. bacteriophora* could be used as a sustainable option for the efficient management of the red palm weevil.

## 1. Introduction

The date palm (*Phoenix dactylifera* L.) belongs to the palm family Arecaceae and is an important fruit crop and economic resource for many countries [1]. It is the oldest fruit crop that has been cultivated since prehistoric times, mostly in arid regions across the globe and has more than 2400 species belonging to 200 genera [2,3,4]. Worldwide, over 100 million date palms are grown on an area of 1 million hectares with a total production of about 7.5 million metric tons of date fruit [5]. Date fruits are considered a complete diet and are a rich source of different nutrients. Dates carry abundant amounts of carbohydrates, dietary fiber, various minerals (calcium, iron, magnesium, phosphorus, potassium, zinc, sulphur, cobalt etc.), proteins, and lipids in trace amounts [6]. The date fruit contains 70% carbohydrates, whereas; date proteins are rich in amino acids [7]. The date fruits also have medicinal value and could play a leading role in preventing the deficiency of vital nutrients in developing countries around the globe [8]. 

The large variety of insect pests that attack various species of palms across the world can be divided into pre- and post-harvest pests. About 112 species of insect pests and mites have been reported to attack the date palm, including 22 species that attack post-harvest stored dates [9,10]. Among these insect and mite pests, the red palm weevil, *Rhynchophorus ferrugineus* (Olivier 1790) (Coleoptera: Curculionidae), is considered the most damaging and invasive pest of the date palm [11]. It was first reported as a damaging pest on Indian coconut palm during 1906 [12]. So far, it has been documented in 54 countries around the globe, damaging more than 40 palm species from 23 palm genera, of which the date, coconut, and Canary Island palms make up a significant proportion [13,14,15,16,17]. 

The adult female of the red palm weevil lays up to 200 eggs at the base of fresh leaves and in wounds in concealed places of the host stem. The neonate grubs bore into the soft fibers and reach the terminal bud tissues, reaching a size of about 5 cm before pupation [18]. The larvae feed on the soft tender tissues of various date palm species. Just before pupation, the full-grown larvae move toward the inner tissues, making tunnels and constructing cocoons from dried fibers. Larval tunneling encourages the infestation of secondary pests and pathogens (e.g., fungal pathogens) [19,20,21]. As the larvae feed on the internal tissues of the palms, the initial infestation is extremely difficult to detect. The infestation is usually noticed after the palm tree has been seriously damaged [22]. The symptoms of infected palms include tunneling in the tree stem from where a yellow fluid oozes out, chewed-up fibers around the holes of tunnels, gnawing sounds made by the growing grubs, pupal cases and adults in the leaf axils, and fallen pupal cases near the base of the date tree. A typical sign of a red palm weevil infestation is a distorted crown in cases of severe damage, which can easily be seen compared with the other symptoms of weevil infestation [23]. Red palm weevils are usually found anywhere within the palm, from the ground surface of the trunk to the apical bud [24]. In the date palm, about 70% of the infestation was found from the ground up to 1–1.5 m, whereas in *Phoenix canariensis*, 80–90% of the infestation was found in the apical portion of the tree [25].

Management of the red palm weevil has presented a significant challenge for entomologists since the start of modern agriculture [26]. Different methods have been employed for the management of red palm weevils, including preventive and curative measures, bioacoustics detection, chemical detection, thermal imaging detection, host plant resistance, phytosanitation and agrotechniques, and integrated pest management programs [14,27,28,29,30,31,32]. However, the main thrust of date palm farmers rely on the repeated and excessive use of synthetic chemicals [33,34]. The indiscriminate use of synthetic chemicals causes many problems, such as the development of resistance in insect pests, accumulation of residues on date palms, deterioration of the environment, and presentation of a serious health hazard to people [35,36]. 

EPNs are a group of soil-dwelling organisms belonging to the Steinernematidae and Heterorhabditidae families. They are well-known natural enemies of insect pests. EPNs play a pivotal role as biocontrol agents of different insect pests, including the red palm weevil [37,38]. The third infective juvenile stage of the EPN is a non-feeding stage that actively looks for target insect pests by utilizing its host-seeking abilities. It invades the host insect and enters the insect body through natural openings such as the spiracles, mouth or anus, or thin parts of the cuticle, and releases a host-specific bacterium (Photorhabdus in the case of Heterorhabditidae; Xenorhabdus in the case of Steinernematidae) that kills the target insect pests through bacterial septicemia [39,40,41]. The entomopathogenic nematodes are easy to culture and inexpensive. They live for weeks during the infective stages and infect a broad range of insect pests [37,40]. The entomopathogenic nematodes have been found to be effective against different insect pests, including the red palm weevil. Several studies have suggested the use of entomopathogenic nematodes as an effective alternative to synthetic insecticides. EPNs are considered host-specific and have no toxic effects on non-target organisms [19,27,42]. They are considered to be most pathogenic against the larvae compared with the other developmental stages of insect pests [43]. There is limited research on the use of entomopathogenic nematodes to control the pupae of the red palm weevil; however, a few studies have been conducted on the efficacy of using entomopathogenic nematodes against the red palm weevil under field conditions. The current investigation tested the relative pathogenicity of four different entomopathogenic nematode species against the multiple developmental stages of the red palm weevil to identify the most susceptible stage for infection and the most effective strain of entomopathogenic nematode. This information would help devise a sustainable integrated management program of the red palm weevil. 

The selected species of nematodes were also tested under natural field conditions to validate their potential as biocontrol agents under more realistic conditions.

## 2. Materials and Methods

### 2.1. Red Palm Weevil Culture

The larvae and adults of red palm weevil were collected from date palm trees at the Germ Plasm Unit (GPU) date palm orchard in Dera Ismail Khan, Khyber Pakhtunkhwa (KPK), Pakistan. The collected stages were shifted to plastic jars (13 × 3 cm^2^) for mass rearing in the laboratory of the Department of Entomology, Gomal University, Dera Ismail Khan (Latitude: 31.8188° N, Longitude: 70.8971° E). The colonies of red palm weevil were maintained in incubators (SANYO Japan, Model MLR-350 H), maintained at constant conditions of 27 ± 2 °C, 65 ± 5% R.H. and 14:10 h D:L regimes.

The collected larvae were provided a soft portion (30 g) of date palm stems (Cultivar: Dhakki) for two days. The jars were washed with tap water every two days and completely air-dried under sunlight. The mouths of the jars were covered with a fine mesh gauze for ventilation purposes. The food for the weevils was changed every alternate day. The full-grown larvae were provided with a five-inch piece of sugarcane to facilitate cocoon formation. The newly formed pupal cocoons were collected and transferred daily to new jars for adult emergence. The emerged adult weevils were cultured in separate jars having a size of 30 × 60 × 60 cm^3^ for feeding, mating, and oviposition purposes. The adult weevils were provided with pieces of the stem, along with sugar and honey solution, to facilitate egg-laying. The red palm weevil adults that emerged on the same day were considered of uniform age and were used for the studies. Newly laid eggs were collected daily to ensure the supply of age-specific red palm weevils for our investigation. 

### 2.2. Entomopathogenic Nematodes (EPNs) 

The initial culture of entomopathogenic nematodes were obtained from the Department of Entomology, the University of Agriculture Faisalabad, Pakistan, and were cultured on the 5th instar larvae of wax moth (*Galleria mellonella*). Nematodes were reared in 9 cm plastic Petri dishes with white filter paper placed on bottom. To maintain the moisture level in the Petri dishes, 1.5 mL tap water along with ≈500 nematodes using a micropipette (MEDLINE SCIENTIFIC LTD., Oxfordshire, UK) were added to each Petri dish. Ten active larvae of wax moth were introduced in each petri dish using a fine camel hairbrush with different species of selected nematodes. The Petri dishes were sealed using a strip of Parafilm and were placed in an incubator maintained at 18 °C for 48 h. After 48 h, the Petri dishes containing wax moth larvae were removed and dead larvae were separated using white traps for the extraction of infective juveniles, following the standard method described by Kaya and Stock [44]. The newly emerged infective juveniles were collected on a daily basis and stored at 18 °C in distilled sterilized water for future use. EPNs less than 5 days old were used for our investigations. 

### 2.3. Laboratory Studies

#### 2.3.1. Pathogenicity of EPNs against Red Palm Weevil Larvae Based on Dissection

A laboratory experiment was conducted to find out the pathogenicity of four EPN species on the 5th instar larvae of the red palm weevil. The experiment was conducted in transparent plastic boxes (9 cm × 5 cm × 5 cm) lined with Whatman No. 1 filter papers. A 0.1 mL solution containing nematodes (approximately 100 IJs) was added using a micropipette to each plastic box, and the solution was allowed to spread evenly on the filter paper. Five 5th instar larvae were collected using fine insect handling forceps and were placed into each box containing nematodes.

The plastic boxes were then sealed and incubated for four days at 20 °C. After the incubation period, the plastic boxes were carefully examined to record data on the mortality of the weevils’ larvae. The larvae infected with nematodes were then dissected under the stereomicroscope (Leica Galen III Professional Microscope (Heerbrugg, Switzerland) in Ringer’s solution. The larvae were carefully observed for the presence of nematodes in the body. The infestation percentage was calculated for statistical analysis. The data on percent mortality were log transformed before analyses.

Using a similar method, the selected nematode strains were tested on the 6th instar larvae of the red palm weevil. The same procedure was used to record mortality data of the 6th instar, except the mortality data was recorded daily until Day 18 and was converted into percent mortality (6th instar). 

#### 2.3.2. Infective Potential of EPNs against Weevil Larvae Based on Adult Emergence

For our second experiment, we used the same procedure as described above using transparent plastic boxes (9 × 5 × 5 cm^3^) with Whatman No. 1 filter paper placed on the bottom and 100 IJs in 0.1 mL water added. Five last instar larvae of the red palm weevil of uniform age were introduced into each plastic box. The experiment was replicated five times for each nematode species. The plastic boxes were sealed using Parafilm to inhibit moisture loss. The plastic boxes were incubated at 20 °C until the last adult emerged. The data were noted on the number of red palm weevil adults that emerged and the nematode species used and was converted into a percent emergence of weevils and nematodes.

#### 2.3.3. Infective Potential of EPNs against Red Palm Weevil Pupae Based on Adult Emergence

The pathogenicity of four selected nematode species, i.e., *H. bacteriophora*, *S. feltiae*, *S. carpocapsae,* and *S. glaseri*, against red palm weevil pupae were evaluated following the procedure described above, with the difference of treating newly formed pupae of the red palm weevil instead of larvae with the selected nematode species. For these investigations, pupae formed within the past 24 h enclosed in their pupal cocoons were used. Five pupae per treatment were added to each plastic box having treated filter paper at the bottom of each box. The pupae were retained in the treated boxes until the emergence of adult weevils. 

Damaged pupae were then dissected under a stereomicroscope in Ringer’s solution and were carefully checked for nematodes in the body of the dead pupae to confirm the cause of pupae mortality. 

### 2.4. Field Studies

The field experiments were conducted in the research area of the Department of Entomology, Gomal University, Dera Ismail Khan, to investigate the infectivity/pathogenicity of four different species of entomopathogenic nematodes (i.e., *H. bacteriophora*, *S. feltiae*, *S. carpocapsae*, and *S. glaseri*) against the red palm weevil. 

For this purpose, date palm plants aged 2 years were used. The date palm plants were grown under lath house conditions to prevent them from insect infestation. The experiment had a randomized complete block design with five repeats, each consisting of five treatments including control. For each treatment, five date palm plants were selected. The date palms were artificially infested with 5 larvae that were 10 days old, in the early morning at 9 a.m. For the release of larvae, a hole 4 cm deep and 1cm wide was drilled into the date palm trunk 1 m above ground level, using an electric drill machine slanted down at a 45⁰ angle. The larvae were released into the trunk and the hole was plugged with cotton covered with mud to inhibit the escape or entry of weevils. Five days following the release of the weevil larvae, another hole of the same size was drilled into the opposite side of the trunk and infested with 5 larvae that were 15 days old, and the hole was concealed after larvae release. Thirty days after the 1^st^ release of the larvae, palm trunks were injected with 6000 infective Juveniles (IJs) nematodes, and the infective capacity of the four selected strains of nematodes was investigated. There were five treatments, comprising of *H. bacteriophora*, *S. carpocapsae*, *S. feltiae*, *S. glaseri*, and control. The palm trunks were dissected 2 weeks after the application of the 1st (IJs) nematode suspension. Data on the number of dead red palm weevils were recorded. To confirm the cause of weevil mortality, all dead weevils were dissected and carefully observed under a microscope (Leica DML 1000 Germany). The data on weevil mortality was converted to % mortality.

### 2.5. Statistical Analysis

The documented data were statistically analyzed using a one-way analysis of variance and means were separated using the least significance difference (LSD) test with α = 0.05. The statistical analysis was carried out using computer software (SPSS ver. 13).

## 3. Results

### 3.1. Pathogenicity of EPNs against Red Palm Weevil Larvae Based on Dissection

The percent infestation of larvae based on dissection was found to be statistically higher in all nematode treatments compared with the control (Table 1). Among the different species of nematodes, *S. carpocapsae* was found to be the most effective, with 94.68% nematode infested larvae, followed by *H. bacteriophora* with 92.68% infestation. There was a significant difference between these two species. The smallest percentages of nematode-infested larvae were found in *S. glaseri* and *S. feltiae* treatments, with 67.60 and 70.88% infestation, respectively. Similarly, all tested nematode species had a significant effect on the percentage of adult emergence of the red palm weevil. Among the nematode species, the lowest adult emergence (4.60%) of weevil was documented in *S. carpocapsae*-treated larvae, showing significantly higher pathogenicity compared with the other tested nematode species. The highest adult emergence in treated weevils was noted in *S. glaseri*-treated larvae. Overall, untreated weevils had the highest adult emergence of 97.80%. 

### 3.2. Infective Potential of EPNs against Red Palm Weevil Pupae Based on Adult Emergence

All of the tested nematode species had significantly higher pathogenicity against the pupae of the red palm weevil compared with the control (Table 2). *S. carpocapsae* was found to be the most effective species, with 63.60% infested pupae, and was significantly different from the other nematode species. *S. glaseri* and *S. feltiae* were the least pathogenic nematodes, having 43.60 and 44.80% infested pupae, respectively. No infested pupae were found in control/untreated pupae. As for adult emergence, the highest adult emergence was noted in *S. glaseri*-treated pupae and *S. feltiae*-treated pupae, with no significant difference between these groups. Among the treatments, the lowest adult emergence of 35.68 and 39.56% was observed in *S. carpocapsae-* and *H. bacteriophora*-treated pupae, respectively. Overall, the highest adult emergence of 98.40% was documented in untreated/control treatment. 

### 3.3. Effect of EPNs on the Mortality of 6th Instar Larvae

The data presented in Figure 1 indicate that all of the tested nematode species were highly pathogenic against the 6th instar larvae of the red palm weevil. The first mortality among treated larvae was noted at 24 h after the exposure period in the *S. carpocapsae*-treated larvae, whereas; *H. bacteriophora* caused the first mortality after 48 h and *S. feltiae* and *S. glaseri* required 72 h to produce the first mortality. The number of weevil mortalities gradually increased with an increased exposure period, with maximum mortality recorded 240 h after the treatment. Among the tested nematode species, *S. carpocapsae* was the most effective treatment, with 100% mortality at 240 h after treatment. This was followed by *H.*
*bacteriophora* with 96% mortality. *S. glaseri* and *S. feltiae* were the least pathogenic nematode species, registering a respective 70 and 76% mortality of 6th instar red palm weevil larvae at 240 h after treatment. No mortality of red palm weevil larvae was documented during the 10-days/240 h exposure period in the control treatment. 

### 3.4. Infective Potential of EPNs against Red Palm Weevil Adults

All of the four evaluated species of nematodes were highly pathogenic against adult weevils (Figure 2). Among the tested nematodes, *S. carpocapsae* was the most pathogenic, followed by *H. bacteriophora*. No mortality of adult weevils was observed in the first three days following treatment. The first mortality was recorded four days after exposure in *S. carpocapsae*-treated adults. *H. bacteriophora* required five days to show pathogenic effects in the adult weevils. *S. feltiae* and *S. glaseri* were the least pathogenic and required six days to cause the first mortality in adult weevils. The highest mortality rate of 70% was observed in adults after 16 days of the treatment in *S. carpocapsae*-treated adults, followed by 63% mortality in adults treated with *H. bacteriophora*. The lowest mortality rates of 59 and 58% were observed in adults treated with *S. feltiae* and *S. glaseri*, respectively, at 16 days after treatment. No further mortality of adult weevils was observed from 16 to 18 days after nematode treatment. 

### 3.5. Infective Potential of EPNs against Red Palm Weevil under Field Conditions

All the tested nematode species were pathogenic against the red palm weevil under field conditions. The *S. carpocapsae* was found to be most effective, causing 83.60% mortality of the red palm weevil. Among the tested four species of nematodes, the *S. glaseri* and *S. feltiae* were the least effective, with 60.60 and 64.80% mortality, respectively. These were statistically significant from each other (Table 3). 

## 4. Discussion

Biological control strategies can yield promising results without deteriorating the quality of the products by managing insect pest populations in different tree species. Entomopathogenic nematodes play a pivotal role as biocontrol agents for different species of insect pests, including the red palm weevil. They actively look for their target by utilizing host-seeking abilities and invade the target by entering the insect body through natural openings. In the current study, entomopathogenic nematodes belonging to the Steinernematidae and Heterorhabditidae families were found to be highly pathogenic against the larvae, pupae, and adult stages of red palm weevils. Among the different species of nematodes, *S. carpocapsae* was the most effective, causing 94.68% nematode-infested larvae, followed by *H. bacteriophora*, with 92.68% infestation. The lowest numbers of nematode-infested larvae were found in *S. glaseri*- and *S. feltiae*-treated larvae. The larval infestation percentage (94.68%) of *S. carpocapsae* was lower than what was reported in Italy [45]. They reported that the *S. carpocapsae* caused 100% mortality of red palm weevil larvae. Similarly, all tested nematode species had a significant effect on the percent adult emergence of the red palm weevil. The minimum adult emergence was documented in *S. carpocapsae-*treated larvae. The results are in agreement with Gözel et al. [46]. They found that the highest mortality (91.4%) of red palm larvae was caused by *S. carpocapsae.* Comparable results were also reported by Llácer et al. [42]. They documented that the nematode *S. carpocapsae* was found highly effective, causing 98% mortality of red palm weevils under field conditions. 

All tested nematode species were less pathogenic against pupae enclosed in their cocoons compared with the larvae and adults of the red palm weevil. However, *S. carpocapsae* was found to be the most effective nematode species with 63.60% infected pupae, followed by *H*. *bacteriophora* with 60.20% infected pupae, whereas; *S. glaseri* and *S. feltiae* were the least pathogenic nematode strains. Atwa and Hegazi [43] reported that EPNs show a preference for infecting red palm weevil larvae over pupae or adults. Similar results regarding the pathogenicity of nematodes against red palm weevils were also recorded by Manzoor et al. [47]. They documented that among the three evaluated nematode species, *S. carpocapsae* were highly pathogenic against the 3rd and 10th larval instar of the red palm weevil, causing 96.5 and 88.17% mortality, respectively. They also reported that *S. feltiae* were least effective, causing only 38.68 and 35.35% mortality of 3rd and 10th instar larvae of the red palm weevil, respectively. Similar findings were also documented by Mikaia [48]. *S. carpocapsae* was the most effective causing the highest mortality of 96.4% of last instar larvae of the red palm weevil under laboratory conditions. In contrast, no significant difference was observed between the pathogenic potential of Heterorhabditids and Steinernematids by Abbas et al. [31]. 

All tested nematode species were highly pathogenic and caused significant mortality in the 6th instar larvae of the red palm weevil. The mortality of the weevils gradually increased with an increase in the exposure period, with maximum mortality recorded after 240 h of treatment. *S. carpocapsae* caused the first mortality among the treated larvae 24 h after the exposure period, whereas; *H. bacteriophora* required 48 h to cause the first mortality of weevils. *S. carpocapsae* and *H. bacteriophora* caused 100 and 96% mortality, respectively, at 240 h/ten days after the application of EPNs. It is evident from the present findings that *S. carpocapsae* and *H. bacteriophora* require ten days to show effectiveness and cause maximum mortality in red palm weevils under laboratory conditions. It can be inferred from these findings that, after the application of EPNs, the treatment requires at least ten days to eradicate/reduce the symptoms of a red palm weevil infestation, including oozing and gnawing sounds. The Sternematids have been reported to be more effective compared with Heterorhabditids against red palm weevil larvae by Santhi et al. [49]. All tested species of nematodes were highly pathogenic against the adult weevils. The highest mortality of 70% was observed in adult weevils after 16 days of treatment in *S. carpocapsae*-treated adults, followed by 63% mortality in adults treated with *H. bacteriophora*. In contrast, the lowest mortality of 59 and 58% was observed in adults treated with *S. feltiae* and *S. glaseri* at 16 days after treatment. Overall, the nematodes were most effective against larvae (5th and 6th instar), followed by adult weevils. Nematodes were the least effective against pupae of the red palm weevil. These results are supported by previous findings reported by Atwa and Hegazi [43]. They reported that different strains of EPNs have different preferences for the various developmental stages of the red palm weevil, and the *Steinernema* sp. was the most effective species compared with other strains of nematodes. Laboratory studies conducted on the susceptibility of the different developmental stages of the red palm weevil to infection by entomopathogenic nematodes have had contradictory results. El Sadaway [50] reported that the Steinernematids were the most pathogenic against the adult stage of the red palm weevil compared with late instar larvae.

All tested nematode species were found equally effective against the red palm weevil under field conditions. Among the tested nematodes, *S. carpocapsae* were found maximally effective, causing 83.60% mortality, whereas; *S. glaseri* and *S. feltiae* were found least effective with 60.60 and 64.80% mortality, respectively. The high mortality of red palm weevil following the application of EPNs under field conditions showed that all of the tested nematode strains survived and had pathogenic potential. The pathogenic potential of *S. carpocapsae* was similar to that reported by Santhi et al. [49]. *S. carpocapsae* was found to be the most virulent nematode strain and caused maximum mortality of 96.4% in the last instar larvae of the red palm weevil. In previous studies, entomopathogenic nematodes have been found to be most pathogenic against the larvae of red palm weevils compared with the pupae or adult weevils, as documented by Cappa et al. [51]. This trend of pathogenicity has also been documented by Williams et al. [52] against the pine weevil, *Hylobius abietis*. Mikia [53] documented that *S. carpocapsae* was found more effective against last instar imago of Brown Marmorated Sting bug *Halyomorha halys* compared to *H. bacteriophora*. The variation in susceptibility according to the different life stages, with larval stages being the most susceptible to infection by nematodes, has also been observed in other insect species [54,55,56]. Our studies confirm that the age and developmental stage of the red palm weevil play a significant role for infection with different strains of entomopathogenic nematodes. Further studies are needed to test the pathogenic potential of these EPNs against the different ages and stages of the red palm weevil under field conditions for various ecological zones of the world. 

## 5. Conclusions

Our results confirmed that *S. carpocapsae* were highly pathogenic compared with the other tested nematode strains, both under laboratory and field conditions, against different developmental stages of the red palm weevil. The larval stage of the red palm weevil is the most susceptible stage for infection with nematodes. We recommend treating of date palm wounds and cracks infected with red palm weevils with live *S. carpocapsae* for the safer management of red palm weevils under field conditions in Pakistan and other regions with similar environmental conditions.

## Figures and Tables

**Figure 1 insects-13-00733-f001:**
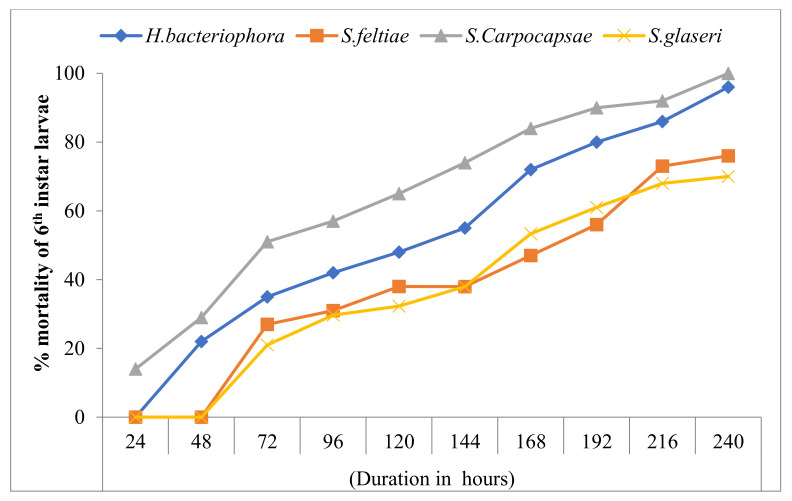
Cumulative mortality of 6th instar larvae of red palm weevil in hours following the application of four species of entomopathogenic nematodes.

**Figure 2 insects-13-00733-f002:**
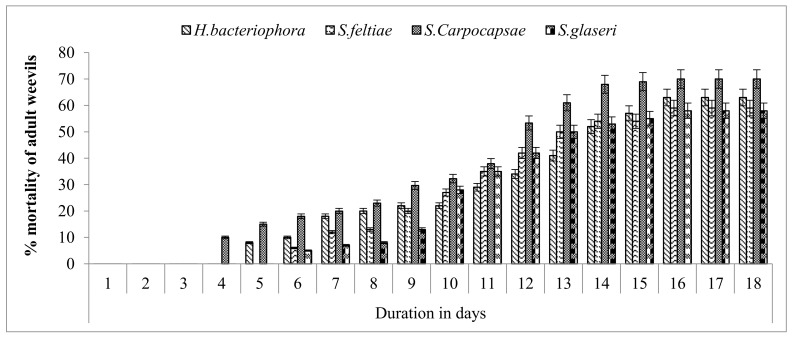
Cumulative mortality of adult red palm weevil in days following the application of four species of entomopathogenic nematodes.

**Table 1 insects-13-00733-t001:** Effect of various species of entomopathogenic nematodes on the % infestation and adult emergence of red palm weevil based on 5th instar larvae treatment.

EPN Species	% Infestation	% Adult Emergence
*H. bacteriophora*	92.68 ± 0.41 b	6.60 ± 0.54 d
*S. feltiae*	70.88 ± 0.29 c	28.60 ± 0.54 c
*S. carpocapsae*	94.68 ± 0.34 a	4.60 ± 0.54 e
*S. glaseri*	67.60 ± 0.37 d	32.40 ± 0.54 b
Control	0.00 ± 0.00 e	97.80 ± 1.13 a
LSD Value	1.34	0.64

Mean ± standard deviation. Means in a column sharing similar letters are statistically similar at 5% level of probability using LSD test.

**Table 2 insects-13-00733-t002:** Percent infestation and adult emergence based on treated pupae of red palm weevil.

EPN Species	% Infestation (Pupae)	% Adults Emerged
*H. bacteriophora*	60.20 ± 0.44 b	39.56 ± 0.51 c
*S.* *feltiae*	44.80 ± 0.83 c	56.20 ± 0.83 b
*S.* *carpocapsae*	63.60 ± 0.54 a	35.68 ± 0.64 d
*S. glaseri*	43.60 ± 1.14 d	56.40 ± 0.54 b
Control	0.20 ± 0.44 e	98.40 ± 0.51 a
LSD	0.96	0.83

Mean ± standard deviation. Means in a column sharing similar letters are statistically similar at 5% level of probability using LSD test.

**Table 3 insects-13-00733-t003:** Effect of four species of entomopathogenic nematodes on the mortality of red palm weevil under field conditions.

EPN Species	% Mortality of Red Palm Weevil
*H. bacteriophora*	80.20 ± 0.44 b
*S. feltiae*	64.80 ± 0.83 c
*S. carpocapsae*	83.60 ± 0.54 a
*S. glaseri*	60.60 ± 1.14 d
Control	0.20 ± 0.44 e
LSD	0.96

Mean ± standard deviation. Means in a column sharing similar letters are statistically similar at 5% level of probability using LSD test.

## Data Availability

The data presented in this manuscript can be made available upon request from the corresponding author.

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
