# Peer review of "Evaluation of Entomopathogenic Nematodes against Red Palm Weevil, Rhynchophorus ferrugineus (Olivier) (Coleoptera: Curculionidae)"

_insects, 2022, doi:10.3390/insects13080733_

Round 1

Reviewer 1 Report

The science is OK and the finding are of value but the writing in terms of grammar and sentence construction needs serious attention 

Author Response

The science is OK and the finding are of value but the writing in terms of grammar and sentence construction needs serious attention 

As per suggestions, the manuscript has been revised comprehensively and all the grammatical and language mistakes has been removed.

Reviewer 2 Report

Hello Dear 

I will attach similar items to this article in one or two cases. And I feel in this regard

Works very close and not very similar to this article have already been published.

https://www.researchgate.net/publication/326843596_Efficacy_of_Entomopathogenic_Nematodes_on_The_Red_Palm_Weevil_Rhynchophorus_ferrugineus_Olivier_1790_Coleoptera_Curculionidae_Larvae

https://www.mdpi.com/2075-4450/13/3/245/htm

Please specify the part that makes this article different, and what is the difference between your work?

Figure one needs a clearer explanation

And grammatically it needs to be changed.

Author Response

Point-1: Please specify the part that makes this article different, and what is the difference between your work?

As no published work was found on the potential of entomopathogenic nematodes against the pupae of red palm weevil, however; a few studies have been conducted on the efficacy of entomopathogenic nematodes against the red palm weevil under field conditions, the current investigations were carried out aiming to figure out the relative pathogenicity of four entomopathogenic nematode species against different developmental stages of red palm weevil

Point-2: Figure one needs a clearer explanation

The necessary change has been made in the revised article Line 254-268.

Point-3: And grammatically it needs to be changed.

The grammatical mistakes have been removed in the revised manuscript

Reviewer 3 Report

1.     The main question I am concerned about for this manuscript is what special result the authors found or the novel idea they would like to show to the reader. As I know, a lot of research about the potential of EPNs to control RPW has been published. They all assessed different EPN species, including the species used here, to the different developmental stages of DRW, and we have already known Sc is a promising species for the controlling. Unsurprisingly, the authors show the same results and conclusions as previous studies. Thus, if the authors want to present the value of this study, they should discuss more concepts or findings from their results, but not show the repeated design of experiments and ideas.

2.     Suggest introducing more content about the application of EPNs for RPW management in the introduction part, to strengthen the description of motivation to conduct this study.

3.     The content of this manuscript is not clear, including the legend of figures without the statement of statistical analysis, the absence of presenting the result in the form of table or figure in 3.1 to 3.3 sections, incorrect legend in figure 2 (also without the title for the y-axis), and inappropriate citation form in the line from 300 to 330.

Author Response

Point-1: The main question I am concerned about for this manuscript is what special result the authors found or the novel idea they would like to show to the reader. As I know, a lot of research about the potential of EPNs to control RPW has been published. They all assessed different EPN species, including the species used here, to the different developmental stages of DRW, and we have already known Sc is a promising species for the controlling. Unsurprisingly, the authors show the same results and conclusions as previous studies. Thus, if the authors want to present the value of this study, they should discuss more concepts or findings from their results, but not show the repeated design of experiments and ideas.

The desired changes have been made in the revised article as suggested by Reviewer-3.  Line 359-364 and Line 329-334.

Point-2: Suggest introducing more content about the application of EPNs for RPW management in the introduction part, to strengthen the description of motivation to conduct this study.

The importance and background about the application of EPNs has been added in the introduction at Line No.100-106.

Point-3:       The content of this manuscript is not clear, including the legend of figures without the statement of statistical analysis, the absence of presenting the result in the form of table or figure in 3.1 to 3.3 sections, incorrect legend in figure 2 (also without the title for the y-axis), and inappropriate citation form in the line from 300 to 330.

All the desired changes have been made in the revised manuscript. Fig-1 and Fig-2 has been replaced with new ones as per suggestions of the reviewer. The legend in Figure-2. i.e., the title of y-axis has been inserted in the new version of the manuscript.

The citation style from Line 300-330 has been corrected.

Reviewer 4 Report

These are my main comments on the MS (insects-1783573):

The introduction and discussion provide no insight on how this MS relates to the various other ones cited in the text or concerns that have been raised by other researchers. This article should provide details on all these fronts to provide the proper context for the work. Authors do not present any hypotheses or expectations that could be connected to previous studies; adding these details will improve the paper. The authors should clearly explain why the research was done, why it was important, and how it fits with other studies. It should be clear and concise, and it is not. The discussion should also include what outcome(s) they expect, and how it would help support or refute their hypotheses or answer their questions (see my comments below). Some of the authors statement would be much stronger if they tie their work to the body of literature that has built up on the invasion ecology and impacts of this pest in urban areas (see Journal of Pest Science, 2019 92(1), 143-156). Statements about insecticide options for palm weevils on lines 80-89 are also missing appropriate references, one suggestion could be: Arthropod Management Tests, 2022 47(1), tsac066; but there are others too.

The M&M section is in poor shape, inappropriate analyses and/or flawed reasoning. ANOVA on percentage mortality data? If so, consider an analysis better suited to this response variable, logistic GLM is one, binomial distribution is another one (see Ecology 2011 92, 3-10). The major difficulty with modelling proportion data is that the responses are strictly bounded. There is no way that the percentage dying can be greater than 100% or less than 0%. But if we use simple techniques such as regression, analysis of variance or covariance, then the fitted model could quite easily predict negative values or values greater than 100%, especially if the variance was high and many of the data were close to 0 or close to 100%. The logistic curve is commonly used to describe data on proportions, because, unlike the straight-line model, it asymptotes at 0 and 1 so that negative proportions and responses of more than 100% cannot be predicted. Briefly, proportions are based on number of cases. Would you give the same weight to a proportion of 2 out of 4 cases (not very reliable) and a more reliable proportion of 20 out of 40 cases? The natural solution is to use the odds and odds ratio, and a binomial distribution to test for change in proportion as a change in the odds, as described in the arcsine asinine publication (see Ecology 2011 92, 3-10). That way you give 50 % of 40 its due, compared to 50% of 4. Results and Discussion sections should be revised accordingly.

The discussion and conclusions lack real concluding remarks in my opinion, and if I was a practitioner or consultant, I’d want to see these recommendations for my area or city. How applicable these findings are to the real world? What are the true benefits of these findings? How would they fit into an existing IPM programs? Adding this information would benefit the discussion.

My minor objections are about the language and awkward phrasing.

I was excited to see the results of the paper after reading the abstract, but I found it hard to extract key messages useful to policymakers and professionals, probably in large part due to the lack of connection with other published work and need for improved structure of the current manuscript. This is not to diminish the data gathered in this study, they are of value. The paper would benefit from a more thorough literature review and a better connection with more relevant reports on the subject.

The next draft of this paper will need to be dramatically different to have a chance at publication in my humble opinion.

Author Response

Point-1: The introduction and discussion provide no insight on how this MS relates to the various other ones cited in the text or concerns that have been raised by other researchers. This article should provide details on all these fronts to provide the proper context for the work. Authors do not present any hypotheses or expectations that could be connected to previous studies; adding these details will improve the paper. The authors should clearly explain why the research was done, why it was important, and how it fits with other studies. It should be clear and concise, and it is not.

The discussion should also include what outcome(s) they expect, and how it would help support or refute their hypotheses or answer their questions (see my comments below). Some of the authors statement would be much stronger if they tie their work to the body of literature that has built up on the invasion ecology and impacts of this pest in urban areas (see Journal of Pest Science, 2019 92(1), 143-156). Statements about insecticide options for palm weevils on lines 80-89 are also missing appropriate references, one suggestion could be: Arthropod Management Tests, 2022 47(1), tsac066; but there are others too.

The introduction has been improved by linking the current research work with the previous studies Line 100-113.

The use of EPN’s for the management of insect pests including red palm weevil has been added and the importance of current studies with gap has been added in the revised manuscript.

The suggestions in the discussion chapter have been incorporated (Line 328-334 and 36-362).

The suggested reference has been added in the revised article (Ref. 51; Line 493-495)

The references regarding the application of insecticides against red palm weevil has been inserted Line 86.

Point-2: The M&M section is in poor shape, inappropriate analyses and/or flawed reasoning. ANOVA on percentage mortality data? If so, consider an analysis better suited to this response variable, logistic GLM is one, binomial distribution is another one (see Ecology 2011 92, 3-10). The major difficulty with modelling proportion data is that the responses are strictly bounded. There is no way that the percentage dying can be greater than 100% or less than 0%. But if we use simple techniques such as regression, analysis of variance or covariance, then the fitted model could quite easily predict negative values or values greater than 100%, especially if the variance was high and many of the data were close to 0 or close to 100%. The logistic curve is commonly used to describe data on proportions, because, unlike the straight-line model, it asymptotes at 0 and 1 so that negative proportions and responses of more than 100% cannot be predicted. Briefly, proportions are based on number of cases. Would you give the same weight to a proportion of 2 out of 4 cases (not very reliable) and a more reliable proportion of 20 out of 40 cases? The natural solution is to use the odds and odds ratio, and a binomial distribution to test for change in proportion as a change in the odds, as described in the arcsine asinine publication (see Ecology 2011 92, 3-10). That way you give 50 % of 40 its due, compared to 50% of 4. Results and Discussion sections should be revised accordingly.

The M&M chapter has been improved in the revised article. The percentage mortality data were log transformed before statistical analyses. As there was no mortality in the control treatment, the control data has been excluded in the revised graph (Fig-1) following the reviewer’s suggestion.

Point-3:       The discussion and conclusions lack real concluding remarks in my opinion, and if I was a practitioner or consultant, I’d want to see these recommendations for my area or city. How applicable these findings are to the real world? What are the true benefits of these findings? How would they fit into an existing IPM programs? Adding this information would benefit the discussion.

The suggestion of the reviewer has been followed in the revised article by refining the discussion and conclusion (Line 379-38).

Point-4:           I was excited to see the results of the paper after reading the abstract, but I found it hard to extract key messages useful to policymakers and professionals, probably in large part due to the lack of connection with other published work and need for improved structure of the current manuscript. This is not to diminish the data gathered in this study, they are of value. The paper would benefit from a more thorough literature review and a better connection with more relevant reports on the subject.

The latest references on the management of red palm weevil have been added by linking the previous findings and discussed in the light of current results.

Round 2

Reviewer 2 Report

Corrections are acceptable.

Reviewer 3 Report

1. I can't agree with the motivation, which should be the most important part, described on lines 105-114. That is totally wrong. Two papers at least which have been published are about the EPNs (including the four species tested here) against different stages of RPW (see attached). Thus, I still have a query regarding the manuscript's originality. 

Atwa, A. A., and Hegazi, E. M. (2014). Comparative Susceptibilities of Different Life Stages of the Red Palm Weevil (Coleoptera: Curculionidae) Treated by Entomopathogenic Nematodes. J. Econ. Entomol. 107, 1339–1347. doi:10.1603/ec13438.

Wakil, W., Yasin, M., and Shapiro-Ilan, D. (2017). Effects of single and combined applications of entomopathogenic fungi and nematodes against Rhynchophorus ferrugineus (Olivier). Sci. Rep. 7, 1–11. doi:10.1038/s41598-017-05615-3.

2. I can't find table 1 and table 3 in the content. This is a big mistake to lead the reader to understand this manuscript. 

3. Typo 'of' in figure 2. 

Reviewer 4 Report

The authors have done a fine job addressing my original comments and suggestions. A few additional remarks have been made below for authors to consider.

L57-59: This statement needs to be referenced, one suggested would-be Journal of Pest Science, 2019 92(1), 143-156

L80-81: Ditto

L81-85: Ditto

A note to the authors out their statistical approach, percentages follow a distribution between 0 and 100; since they are calculated as a ratio, the distribution is horrible. It is certainly possible to change a percentage to a log if there are no 0%’s. But is that what you really want? There is no reason to transform the data to logs. Maybe for a future reference you should think of a logit or binomial model instead. This paper is a useful start http://www3.nd.edu/~rwilliam/stats3/FractionalResponseModels.pdf

Round 3

Reviewer 3 Report

Accept in present form.